# Keratin 14-dependent disulfides regulate epidermal homeostasis and barrier function via 14-3-3σ and YAP1

Yajuan Guo[1,2], Catherine J Redmond[2], Krystynne A Leacock[1], Margarita V Brovkina[3], Suyun Ji[2], Vinod Jaskula-Ranga[4], Pierre A Coulombe[1,2,5,6]*

[1]Department of Biochemistry and Molecular Biology, Johns Hopkins Bloomberg School of Public Health, Baltimore, United States; [2]Department of Cell and Developmental Biology, University of Michigan Medical School, Ann Arbor, United States; [3]Graduate Program in Cellular and Molecular Biology, University of Michigan Medical School, Ann Arbor, United States; [4]Department of Ophthalmology, Johns Hopkins School of Medicine, Baltimore, United States; [5]Department of Dermatology, University of Michigan Medical School, Ann Arbor, United States; [6]Rogel Cancer Center, Michigan Medicine, University of Michigan, Ann Arbor, United States

**Abstract** The intermediate filament protein keratin 14 (K14) provides vital structural support in basal keratinocytes of epidermis. Recent studies evidenced a role for K14-dependent disulfide bonding in the organization and dynamics of keratin IFs in skin keratinocytes. Here we report that knock-in mice harboring a cysteine-to-alanine substitution at *Krt14*'s codon 373 (C373A) exhibit alterations in disulfide-bonded K14 species and a barrier defect secondary to enhanced proliferation, faster transit time and altered differentiation in epidermis. A proteomics screen identified 14-3-3 as K14 interacting proteins. Follow-up studies showed that YAP1, a transcriptional effector of Hippo signaling regulated by 14-3-3sigma in skin keratinocytes, shows aberrant subcellular partitioning and function in differentiating *Krt14* C373A keratinocytes. Residue C373 in K14, which is conserved in a subset of keratins, is revealed as a novel regulator of keratin organization and YAP function in early differentiating keratinocytes, with an impact on cell mechanics, homeostasis and barrier function in epidermis.

*For correspondence:
coulombe@umich.edu

Competing interests: The authors declare that no competing interests exist.

## Introduction

The epidermis covering our skin and body maintains a vital and multidimensional barrier to water and the outside environment while renewing itself with rapid kinetics, even under normal physiological conditions (*Kubo et al., 2012*). The mechanisms through which new progenitor cells are produced at the base of this stratified epithelium, pace themselves through differentiation, and maintain tissue architecture and function in spite of a high rate of cell loss at the skin surface are only partially understood (*Wells and Watt, 2018*).

Keratin intermediate filaments are major protein constituents in epithelial cells and are encoded by a large family of 54 conserved genes that are individually regulated in a tissue- and differentiation-specific fashion (*Schweizer et al., 2006*). An outstanding question is the extent to which keratin, and other types of intermediate filaments (IFs), participate in basic processes such as cell differentiation and tissue homeostasis. The type I keratin 14 (K14) and type II K5 co-polymerize to form the prominent IF apparatus that occurs in the progenitor basal layer of epidermis and related complex epithelia (*Nelson and Sun, 1983*; *Fuchs, 1995*). Two main roles have so far been ascribed to K5-K14

IFs. First, to provide structural support and mechanical resilience to keratinocytes in the basal layer of epidermis and related epithelia (*Coulombe et al., 1991a*; *Vassar et al., 1991*; *Fuchs and Coulombe, 1992*). Second, to regulate the distribution of melanin with an impact on skin pigmentation and tone (*Uttam et al., 1996*; *Betz et al., 2006*; *Gu and Coulombe, 2007*). Dominantly-acting missense alleles in either *KRT5* or *KRT14* underlie the vast majority of cases of epidermolysis bullosa simplex (EBS), a rare genetic skin disorder in which trivial trauma results in skin blistering secondary to the lysis of fragile basal keratinocytes (*Bonifas et al., 1991*; *Coulombe et al., 1991b*; *Fuchs and Coulombe, 1992*; *Lane et al., 1992*). Such mutant alleles may also affect skin pigmentation (*Gu and Coulombe, 2007*), establishing the relevance of both roles of K5-K14 in both healthy and diseased skin.

Structural insight gained from solving the crystal structure of the interacting 2B regions of corresponding rod domain segments in human K5 and K14 highlighted the presence of a trans-dimer, homotypic disulfide bond involving cysteine (C) residue 367 (C367) in K14 (*Coulombe and Lee, 2012*). Conspicuously, residue C367 in K14 occurs within a four-residue interruption, or stutter, in the long-range heptad repeat of coil two in the central alpha-helical rod domain in virtually all IF proteins (*Lee et al., 2012*). We showed that K14 C367-dependent disulfides form in human and mouse skin keratinocytes (*Lee et al., 2012*), where they play a role in the assembly, organization and steady state dynamics of keratin IFs in live skin keratinocytes (*Feng and Coulombe, 2015a*; *Feng and Coulombe, 2015b*). We also showed that loss of the stutter cysteine alters K14's ability to become part of the dense meshwork of keratin filaments that occurs in the perinuclear space of early differentiating keratinocytes (*Lee et al., 2012*; *Feng and Coulombe, 2015a*; *Feng and Coulombe, 2015b*). However, the physiological significance associated with the surprising properties conferred by a cysteine residue located in a mysterious conserved motif within the central rod domain of a keratin, namely K14, remained unclear.

Here, we report on studies involving a new mouse model that provides evidence that the stutter cysteine in K14 protein regulates entry into differentiation and thus the balance between proliferation and differentiation through regulated interactions with 14-3-3 adaptor proteins and YAP1, a terminal effector of Hippo signaling (*Pocaterra et al., 2020*). We also discuss evidence that this role likely applies to K10 and other type I keratins expressed in surface epithelia.

## Results

The distribution of cysteine residues in mouse K14 protein is schematized in *Figure 1A*. Codon C367 in *KRT14* (human) occurs at position 373 in *Krt14* (mouse), and is conserved in the orthologous keratin of several other species (*Figure 1B*). Moreover, this codon is also conserved in many other type I keratin genes expressed in skin (*Strnad et al., 2011*; *Lee et al., 2012*; *Figure 1B*). To address the physiological significance of the conserved stutter cysteine in K14, we generated *Krt14* C373A mutant mice using CRISPR-Cas9 technology (*Figure 1C*) and verified its presence through allele specific DNA-sequencing (*Figure 1D*). *Krt14* C373A mice are born in the expected mendelian ratio, are viable and fertile, and show a normal body weight when reaching adulthood (*Figure 1E*). Analysis of total skin proteins from several body sites showed that steady state levels of K14 protein are unaffected in *Krt14* C373A relative to WT skin. By contrast, the pattern of K14-dependent, high molecular weight disulfide-bonded species is markedly altered, given fewer species that occur at lower levels (*Figure 1F,G*). This is so especially in ear and tail skin (*Figure 1F,G*), prompting us to focus on these two body sites in subsequent analyses. The residual K14-dependent disulfide bonding occurring in *Krt14* C373A mutant skin (*Figure 1F,G*) likely reflects the participation of cysteines located in the N-terminal domain of K14 (see *Figure 1A* and *Feng and Coulombe, 2015a*). These findings indicate that mice homozygous for *Krt14* C373A allele are viable and appear macroscopically normal, although biochemically they exhibit a strikingly altered pattern of K14-dependent disulfide bonding, particularly in ear and tail skin.

The histology and barrier status of young adult *Krt14* C373A mouse skin were analyzed next. By histology, the epidermis of *Krt14* C373A mice is modestly but significantly thickened relative to WT in ear and tail skin (*Figure 2A,B*). Measurement of trans-epidermal water loss (TEWL) at the skin surface revealed an increase in *Krt14* C373A mice relative to WT control. This is so both at baseline ($7.02 \pm 0.72$ g/m2/h vs. $2.95 \pm 0.49$ g/m$^2$/h) and after topical acetone application ($19.20 \pm 1.78$ g/m2/h vs. $7.02 \pm 0.72$ g/m$^2$/h) (*Figure 2C*), a standard challenge that puts the skin barrier under a

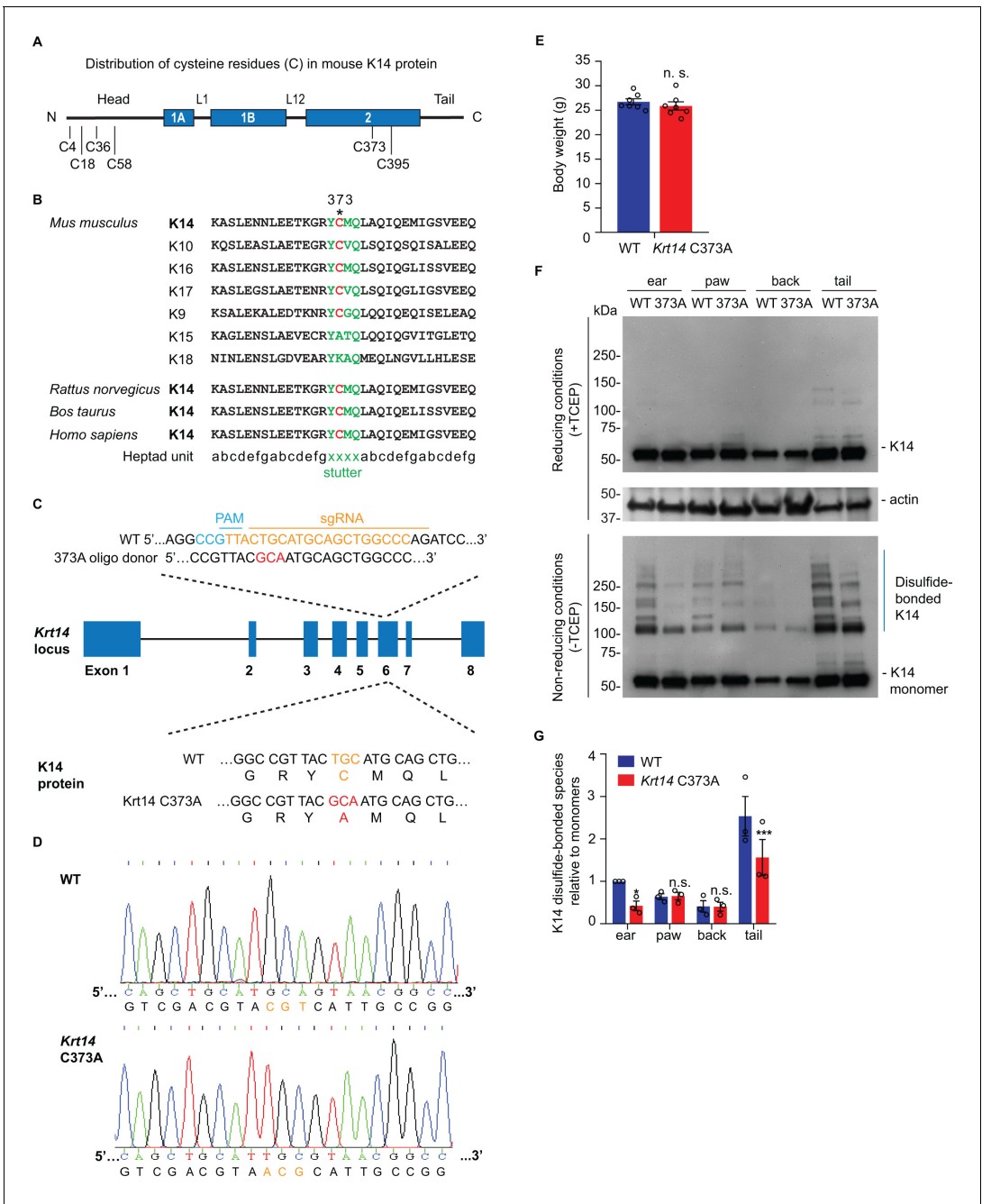

**Figure 1.** Decreased K14-dependent disulfide-bonded species and thickened epidermis in *Krt14* C373A mouse skin. (**A**) Location of cysteine (C) residues in mouse K14 protein (C4, C18, C36, C58, C373, C395), in which N-terminal head and C-terminal tail domains are flanking the central α-helical rod domain (coils 1A, 1B and 2 (blue boxes) separated by linkers L1 and L12). (**B**) Alignment of the sequence context flanking residue C373 in mouse K14 and other mouse type I keratins (top) as well as for K14 in other species (bottom). The heptad repeat is shown at the bottom. 'xxxx' marks the location of the stutter sequence (green letters). (**C**) Schematic diagram of the strategy used to generate *Krt14* C373A mice using the Crispr/Cas9 system. sgRNA, single guide RNA; PAM, protospacer adjacent motif. (**D**) Sanger sequencing showing the TGC to GCA transversion at codon 373 (cysteine to alanine) in the *Krt14* gene. (**E**) Young adult *Krt14* C373A WT littermate male mice show a similar body mass. N = 7 for each genotype. (**F**) Immunoblotting analysis of total protein lysates from ear, paw, back skin, and tail skin from WT and *Krt14* C373A young adult mice subjected to SDS-PAGE electrophoresis under reducing (+TCEP) and non-reducing (-TCEP) conditions. (**G**) Quantification of relative amounts of K14-dependent disulfides over monomers (see c). N = 3 replicates. Data represent mean ± SEM. Student's t test: n.s., no statistical difference; *p<0.05; ***p<0.005.

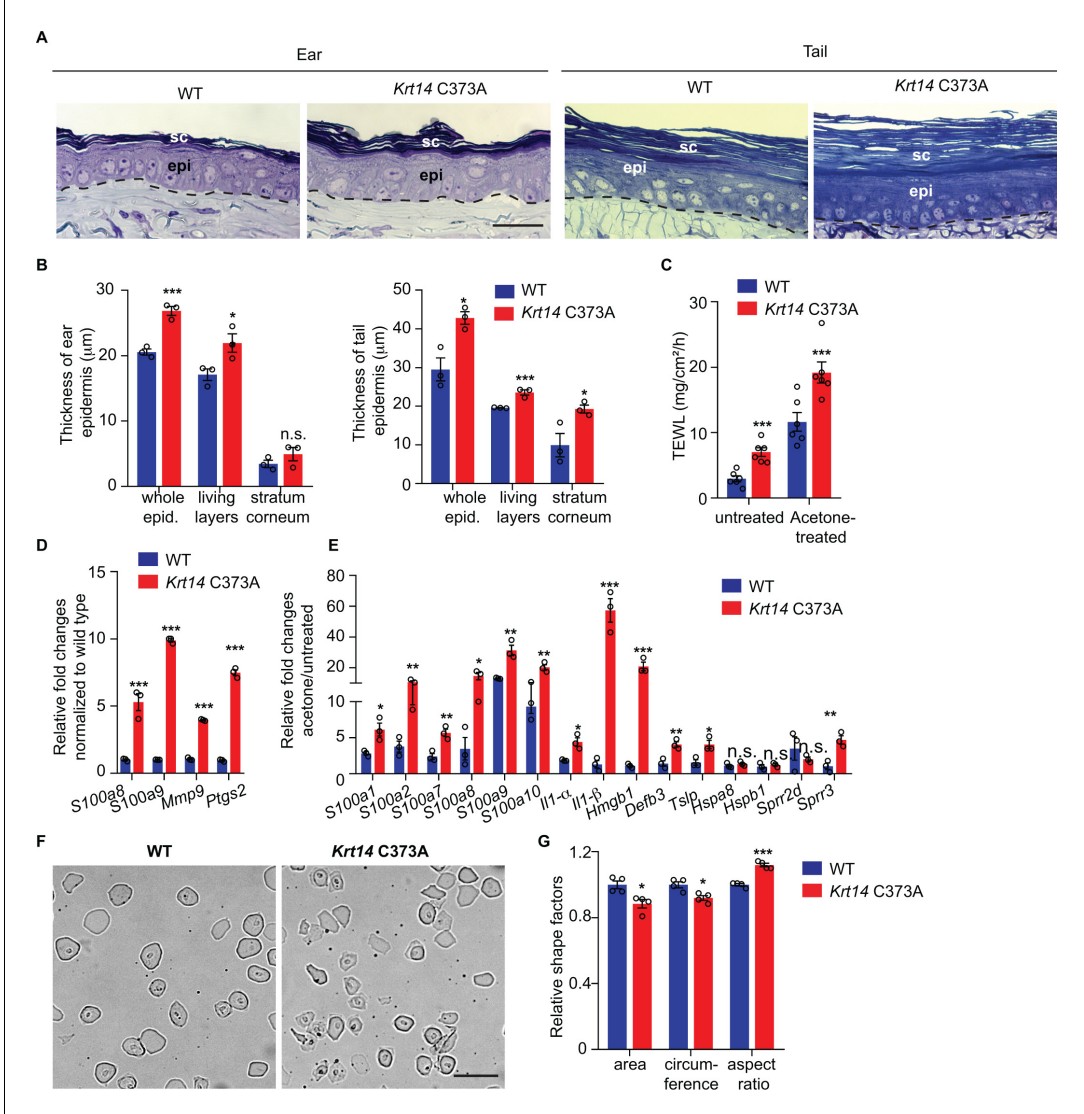

**Figure 2.** Alternations in morphology and barrier status in *Krt14* C373A skin. (**A**) Toluidine blue-stained sections (1 mm thick from epoxy-embedded skin of young adult WT and *Krt14* C373A mice. (**B**) Quantification of whole epidermal thickness (living epidermal layers and stratum corneum layers) in ear (left) and tail (right) skin of WT and *Krt14* C373A mice. Five random fields were sampled for each of 3 mice per genotype. Scale bar, 20 μm. (**C**) Trans-epidermal water loss measurements of WT and *Krt14* C373A ear skin at baseline (untreated) and after acetone-induced barrier disruption. N = 6 per sample. D. Relative fold change in mRNA levels (qRT-PCR) for Danger-Associated Molecular Patters (DAMPs) in WT and *Krt14* C373A skin at baseline. N = 3 biological replicates. (**E**) Relative fold change in mRNA levels (qRT-PCR) for DAMPs after acetone treatment. N = 3 biological replicates. (**F**) Representative phase contrast microscopy images of cornified envelopes isolated from WT and *Krt14* C373A tail skin. (**G**) Quantitation of surface area, circumference, and aspect ratio of cornified envelopes in d. Approximately 100 CEs were counted for each of four mice. Data represent mean ± SEM. Student's t test: *p<0.05; **p<0.01; ***p<0.005; n.s., no statistical difference. Scale bar, 100 μm.

The online version of this article includes the following figure supplement(s) for figure 2:

**Figure supplement 1.** Analysis of purified cornified envelopes from ear skin.

mild and reversible stress (*Denda et al., 1996*). Skin barrier defects often trigger elevated expression of Danger-Associated Molecular Patterns (*Lessard et al., 2013*) (DAMPs, also known as alarmins). At baseline, DAMPs such as *S100a8*, *S100a9*, *Mmp9*, and *Ptgs2* are upregulated by 4-fold or more at the mRNA level in *Krt14* C373A skin compared to WT (*Figure 2D*). This aberrant state is markedly enhanced after a topical acetone challenge to the barrier (*Figure 2E*). Next, we analyzed cornified envelopes (CEs) isolated from epidermis, given that they are key contributors to skin barrier function (*Eckhart et al., 2013*). CEs harvested from WT mice appear relatively uniform in size and

shape, are mostly oval-shaped, and feature clear and smooth outlines (*Figure 2F,G*). By contrast, CEs isolated from *Krt14* C373A mice are smaller (85% of the area and 87% of the circumference of WT CEs), jagged, and less oval-shaped (aspect ratio of 1.3 compared to 1.1 in WT) (*Figure 2F,G* and *Figure 2—figure supplement 1A,B*). Thus, the morphological and molecular anomalies occurring in the epidermis are accompanied by significant defects in barrier function in *Krt14* C373A skin.

We next assessed keratinocyte proliferation, transit time, and apoptosis in order to identify possible causes for the increased thickness and barrier defect in *Krt14* C373A epidermis. At 2 hr after a single pulse of the nucleotide analog Edu (*Chehrehasa et al., 2009*), a significantly greater fraction of keratinocytes are labeled in the basal layer of *Krt14* C373A epidermis compared to WT (by ~1.6 fold; p=0.024) (*Figure 3A,B*), indicating that keratinocyte proliferation is enhanced at baseline in mutant mice. Following a 1 day chase after the Edu pulse, this difference is accentuated (>2 fold; p=0.01) and Edu-labeled nuclei now occur in the suprabasal layers of epidermis in both genotypes, reflecting keratinocyte exit from the basal layer (*Figure 3A,B*). Following a 3-day chase after the Edu pulse, nuclear labeling remains high and stable in the basal layer of epidermis in both genotypes, but a clear additional difference emerges as there are significantly more Edu-labeled nuclei in the suprabasal layers of mutant epidermis. At the 7-day mark, the fraction of labeled cells in the basal layer has subsided in both genotypes but, again, the suprabasal epidermis of *Krt14* C373A skin shows far more labeled nuclei (*Figure 3A,B*). This pulse-chase experiment shows that keratinocytes in *Krt14* C373A epidermis show enhanced proliferation confined to the basal layer at baseline, and that keratinocytes exhibit a faster pace of movement across the suprabasal layers as they progress through differentiation. A similar phenotype has been previously described for *Krt10* null mice (*Reichelt and Magin, 2002*). We also observed a greater frequency (~3 fold) of TUNEL-positive nuclei in *Krt14* C373A epidermis compared to WT, with apoptotic cell death confined to the suprabasal compartment (*Figure 3C,D*). We also examined expression of p63, given its role as master regulator of epidermal stratification and differentiation (*Soares and Zhou, 2018*). A distinct immunostaining pattern was observed in *Krt14* C373A epidermis relative to WT (*Figure 3E*). Upon quantitation, significant differences prevailed in terms of frequency of keratinocytes labeled and their distance from the basal lamina (*Figure 3F*), with p63-positive staining showing a conspicuous elongated shape and extending higher up in the mutant epidermis. When combined, these findings suggest that the modest increase observed in epidermal thickness (*Figure 1*) masks a more pronounced defect in epidermal homeostasis under baseline conditions in *Krt14* C373A mice.

We next assessed markers relevant to keratinocyte proliferation to identify possible causes for the defective barrier of *Krt14* C373A skin relative to WT. We examined the distribution of K14 (basal cell layer), K10 (early differentiation), filaggrin and loricrin (late differentiation) in tail skin sections from young adult mice. The staining for filaggrin and loricrin were markedly decreased (~62% and 48% reductions, respectively) while the staining for K10 was modestly decreased (~37% reduction) in *Krt14* C373A epidermis relative to WT (*Figure 3G,H*). We also examined markers of adherens junction (E-cadherin), desmosomes (desmoplakin), tight junctions (claudin 3) since epidermal differentiation entails a tightly coordinated rearrangement of intercellular junctions. Claudin 3 staining was decreased by ~33% in *Krt14* C373A epidermis, consistent with the barrier defect. The signals for E-cadherin and desmoplakin appeared slightly increased (*Figure 3I*), an occurrence that may reflect the modest epidermal thickening (*Figure 3J*). In contrast to tail and ear epidermis, several markers including K14, K10, loricrin and filaggrin appear normal in the thin epidermis of back skin (*Figure 3—figure supplement 1A–C*), consistent with the markedly lower yield of K14-dependent disulfide bonding in this body site (*Figure 1*). Together these observations link the anomalies observed in epidermal homeostasis and skin barrier to defects in terminal keratinocyte differentiation in *Krt14* C373A mouse skin.

We previously showed that replacing Cys with Ala at position 367 in human K14 does not abrogate 10 nm filament formation but leads to a reduction in the perinuclear clustering of keratin filaments in cultured keratinocytes (*Feng and Coulombe, 2015b*). Transmission electron microscopy of epoxy-embedded skin tissue sections was next used to assess whether similar changes occur in vivo. In basal keratinocytes of WT epidermis, keratin IFs typically occur as bundles near the nucleus. In *Krt14* C373A basal keratinocytes, however, keratin IFs are absent from the perinuclear region and appear redistributed towards the cell periphery (*Figure 3—figure supplement 2A,B*). Consistent with the macroscopic appearance of skin tissue there is no ultrastructural evidence of cell fragility in *Krt14* C373A epidermis (*Figure 3—figure supplement 2A,B* and data not shown). We find that

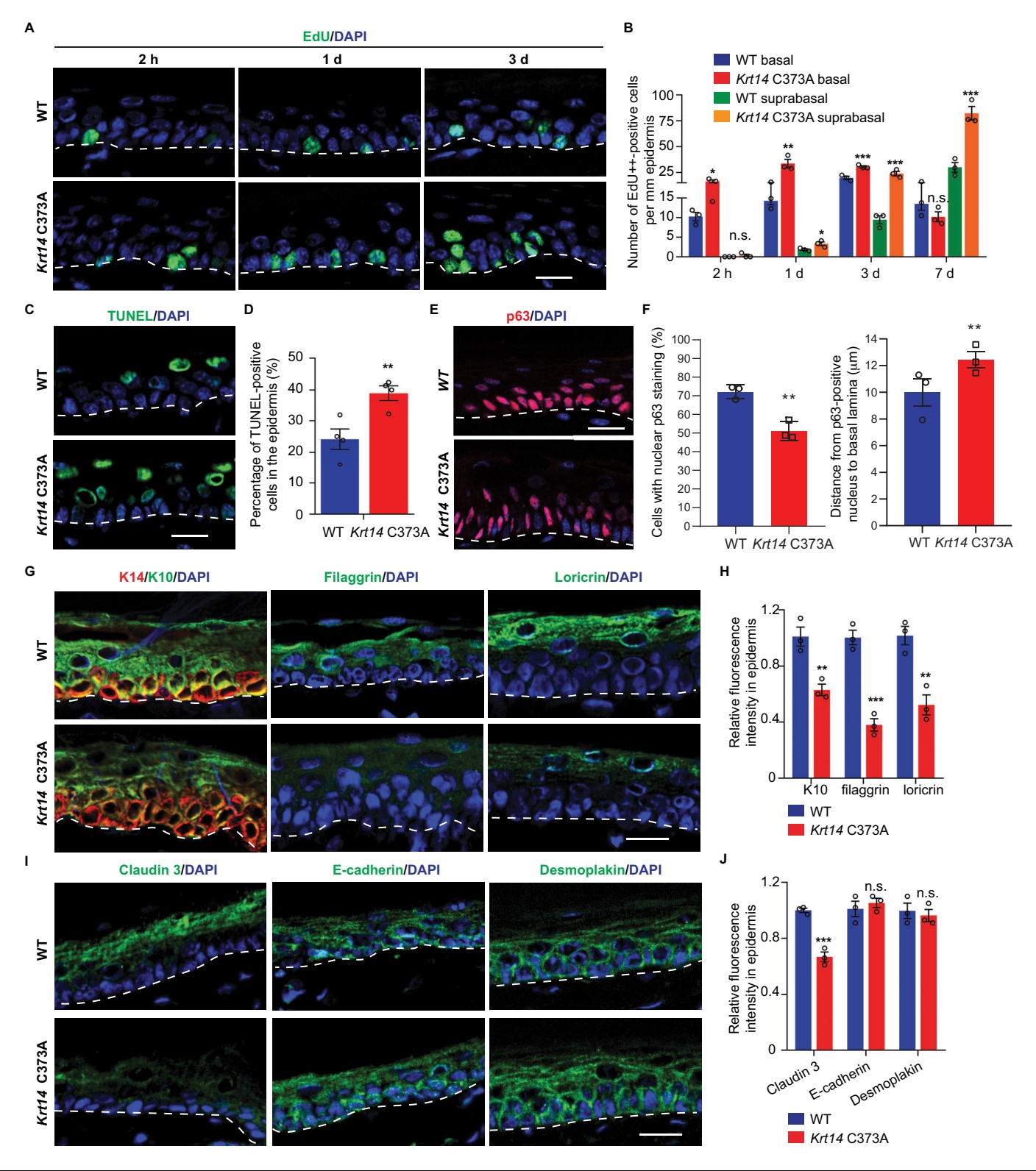

**Figure 3.** Altered tissue homeostasis and dysregulated keratinocyte differentiation in *Krt14* C373A skin. (**A**) Indirect immunofluorescence for Edu in tail skin section from WT and *Krt14* C373A at 2 hr, 1 d, and 3 d after treatment with thymidine analog EdU. Nuclei as stained with DAPI (blue). (**B**) Quantification of number of EdU-positive nuclei in basal and suprabasal layers per mm of epidermis. N = 3 replicates for each sample. (**C**) TUNEL staining in tail epidermis of young adult WT and *Krt14* C373A mice. D. Quantification of TUNEL-positive cells shown in frame c. N = 4 mice per sample. *Figure 3 continued on next page*

*Figure 3 continued*

E. Indirect immunofluorescence for p63 in tail skin section from WT and *Krt14* C373A tail skin. Dashed lines depict the dermo-epidermal interface. (F) Quantification of the number of p63-positive nuclei per mm of epidermis (left) and their distance from the basal lamina (right). N = 3 replicates for each sample. (G) Indirect immunofluorescence for K14 (green), K10 (red), filaggrin, and loricrin from tail skin sections of WT and *Krt14* C373A mice. (H) Quantification of relative fluorescence intensity of data shown in frame g, normalized to WT. N = 3 mice per sample. (I) Indirect immunofluorescence for claudin 3, E-cadherin and desmoplakin in tail skin sections from WT and *Krt14* C373A mice. (J) Quantitation of relative fluorescence intensity in g. N = 3 mice per sample. In a, c, e, g, and I, nuclei are stained with DAPI (blue), and dashed lines depict the dermo-epidermal interface. Scale bars, 20 μm. Data in b, d, f, h and g represent mean ± SEM. Student's t test: *p<0.05; **p<0.01; ***p<0.005; n.s., no difference.

The online version of this article includes the following figure supplement(s) for figure 3:

**Figure supplement 1.** Analysis of terminal differentiation in mouse back skin tissue.
**Figure supplement 2.** Ultrastructural changes and abnormal nuclei in *Krt14* C373A keratinocytes.

nuclei feature a more ellipsoid shape along with a greater frequency of cytoplasmic invaginations (by ~1.4 fold in basal keratinocytes and by ~1.7 fold in suprabasal keratinocytes, respectively) compared to WT controls; *Figure 3—figure supplement 2C*). The occurrence of ultrastructural anomalies in the perinuclear keratin IF network in *Krt14* C373A basal keratinocytes extend previous live imaging observations (*Feng and Coulombe, 2015b*) and point to the possibility that the mechanical properties of the nuclear envelope or nucleus are altered in these cells.

To identify potential pathways regulated by K14-dependent disulfides, we performed K14 co-immunoprecipitation (co-IP) assays followed by mass spectrometry (MS) analysis in protein extracts prepared from newborn WT keratinocytes in primary culture in the presence of 1 mM $Ca^{2+}$, a condition that induces keratinocyte differentiation (*Hennings et al., 1980*). This screen identified 14-3-3σ and other 14-4-3 isoforms as major interacting partners for K14 in WT cell cultures (*Figure 4A* and *Supplementary file 1*). There is a strong precedent for interactions between 14-3-3 proteins and keratins, including K18 (*Liao and Omary, 1996*; *Ku et al., 1998*), K17 (*Kim et al., 2006*), which occur in a phosphorylation-dependent fashion. Co-immunoprecipitation assays confirmed that transfected, HA tagged-14-3-3σ physically interact with endogenous WT K14 and *Krt14*373A mutant protein in mouse keratinocytes in primary culture (*Figure 4B* and data not shown). Further inspection of the top 100 MS-identified proteins in this targeted proteomics screen (*Supplementary file 1*) reveals, as expected, the presence of desmosomal proteins, known keratin-interacting proteins (e.g., annexins; (*Chung et al., 2012*), and several proteins with known roles in organelle transport and organization (e.g., rab family members; *Ohbayashi and Fukuda, 2012*), which is consistent with K5-K14's established role in skin pigmentation (*Gu and Coulombe, 2007*).

14-3-3σ was deemed of interest because it regulates the proliferation and differentiation of keratinocytes in epidermis (*Herron et al., 2005*; *Li et al., 2005*). The latter is achieved in part by modulating the cellular localization of YAP (*Li et al., 2005*; *Sun et al., 2015*), a terminal effector of Hippo signaling (*Schlegelmilch et al., 2011*; *Silvis et al., 2011*; *Sambandam et al., 2015*). Hippo is an evolutionary conserved pathway with a primary role in regulating growth and homeostasis in organs and tissues (*Pocaterra et al., 2020*). We next assessed the distribution of 14-3-3σ and YAP using indirect immunofluorescence of tissue sections prepared from WT and *Krt14* C373A tail skin. 14-3-3σ occurs mostly as aggregates in suprabasal keratinocytes of *Krt14* C373A epidermis, which is in striking contrast to the diffuse distribution observed in WT controls (*Figure 4C*). Consistent with previous reports (*Schlegelmilch et al., 2011*; *Sambandam et al., 2015*), a strong signal for YAP occurs in both the nucleus and cytoplasm in basal keratinocytes, and otherwise YAP occurs as a weaker and diffuse signal in the cytoplasm (but is not seen in the nucleus) of suprabasal keratinocytes in WT epidermis (*Figure 4D*). In *Krt14* C373A epidermis, strikingly, YAP localizes preferentially to nuclei in both basal and suprabasal keratinocytes, in a consistent fashion (*Figure 4D*). The latter finding suggests that YAP-dependent gene expression may be altered in mutant mouse skin. Follow-up RT-qPCR assays show that the steady state levels for several known YAP target gene mRNAs, including *Cyr61*, *Zeb1*, *Ctgf* and *Snail2*, are markedly elevated in *Krt14* C373A relative to WT skin (*Figure 4E*). By western immunoblotting, the levels of endogenous YAP1 and Ser[127]-phosphorylated YAP1 are similar in WT and *Krt14* C373A skin (*Figure 4F,G*). Together these findings point to a misregulation of 14-3-3σ and YAP as likely contributors to the epidermal phenotype exhibited in the ear and tail skin of *Krt14* C373A mice.

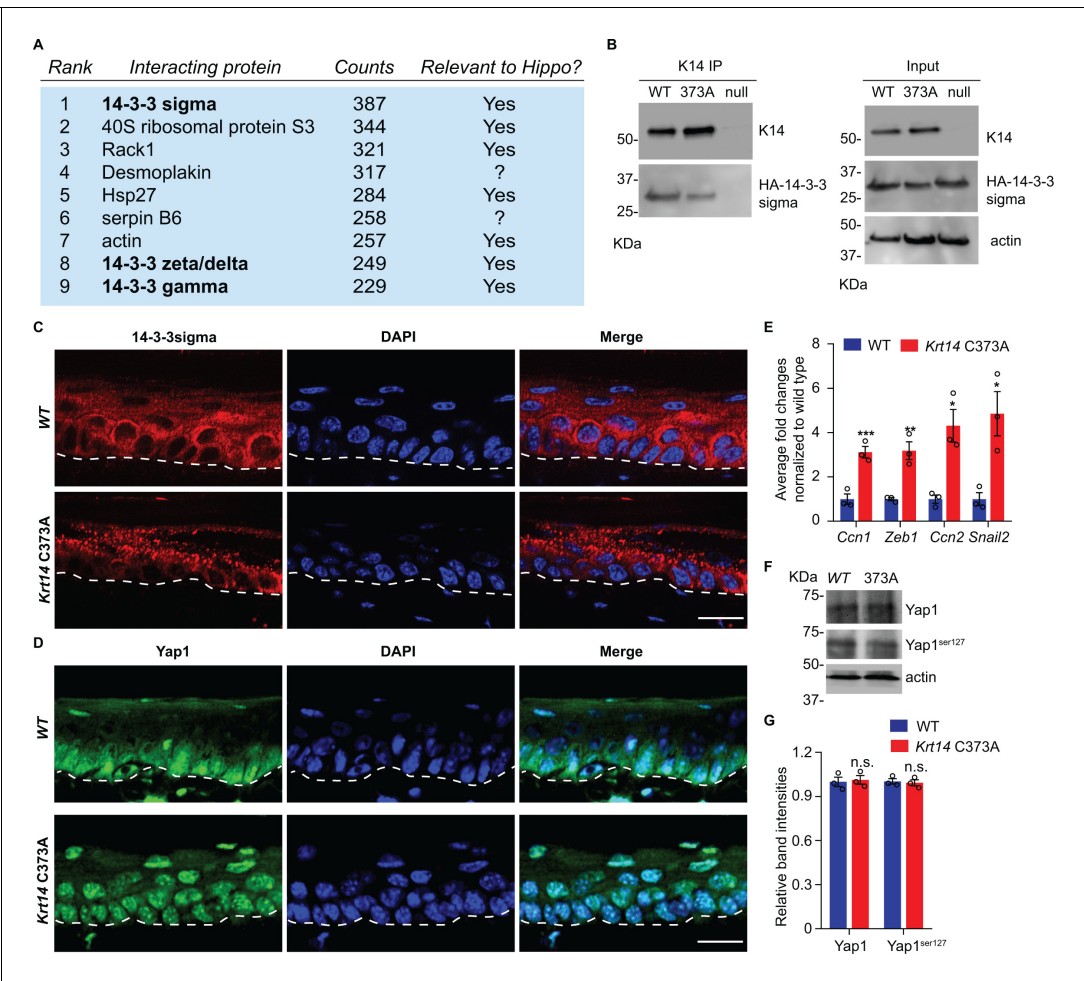

**Figure 4.** 14-3-3σ interacts with K14, and abnormal localization 14-3-3σ and YAP in *Krt14* C373A epidermis. (**A**) Top nine most abundant non-keratin entries from a mass spectrometry screen for proteins interacting with K14 in WT newborn skin keratinocytes (primary culture,1 mM calcium, 4 days). Spectral counts and known relevance to Hippo signaling are indicated. See *Figure 4—figure supplement 1* for full listing. (**B**) Immunoprecipitation of K14 from WT or *Krt14* C373A skin keratinocytes in primary culture. Both K14 WT and, albeit to a lesser extent, the 373A mutant interact with HA-tagged 14-3-3σ. KDa, kilodalton. (**C**) Indirect immunofluorescence for 14-3-3σ in WT and *Krt14* C373A tail skin sections. Dashed lines depict the dermo-epidermal interface. (**D**) Indirect immunofluorescence for YAP in WT and *Krt14* C373A tail skin sections. (**E**) Relative mRNA levels (qRT-PCR) for YAP target genes *Ccn1*, *Zeb1*, *Ccn2*, and *Snail2* in adult WT and *Krt14* C373A tail skin. N = 3 biological replicates per genotype. (**F**) Immunoblotting analysis for total YAP and YAP$^{Ser127}$ in WT and *Krt14* C373A tail skin protein lysates. (**G**) Quantification of relative protein levels shown in frame d. Data are mean ± SEM from three biological replicates. Student's t test: *p<0.05; **p<0.01; ***p<0.005; n.s., no statistical difference. Scale bars, 20 μm.

The online version of this article includes the following figure supplement(s) for figure 4:

**Figure supplement 1.** Additional analyses of newborn skin keratinocytes in primary culture.

Next we asked whether the misregulation of YAP subcellular partitioning also occurs in primary culture. Keratinocytes were isolated from WT and *Krt14* C373A newborn pups, cultured in the absence or presence of calcium (*Hennings et al., 1980*), and analyzed using microscopy-based read-outs. K14-dependent disulfide bonding is low in the absence of calcium and rises of the course of days after adding calcium to primary cultures of WT mouse keratinocytes (*Figure 4—figure supplement 1A,B*). In the absence of calcium, the staining for YAP is concentrated in the nucleus in both WT and *Krt14* C373A keratinocytes (*Figure 5A,B*). After addition of calcium (1 mM), which triggers differentiation and mimics a suprabasal state (*Hennings et al., 1980*), 73% of WT keratinocytes lose their nuclear YAP signal whereas 95% of *Krt14* C373A keratinocytes exhibit predominantly nuclear YAP (*Figure 5A,B*). Western immunoblotting confirmed that, as expected, both K10 and filaggrin proteins occur at lower levels in calcium-treated primary cultures of *Krt14* C373A relative to WT keratinocytes (*Figure 4—figure supplement 1C,D*), suggesting a delay or a defect in terminal

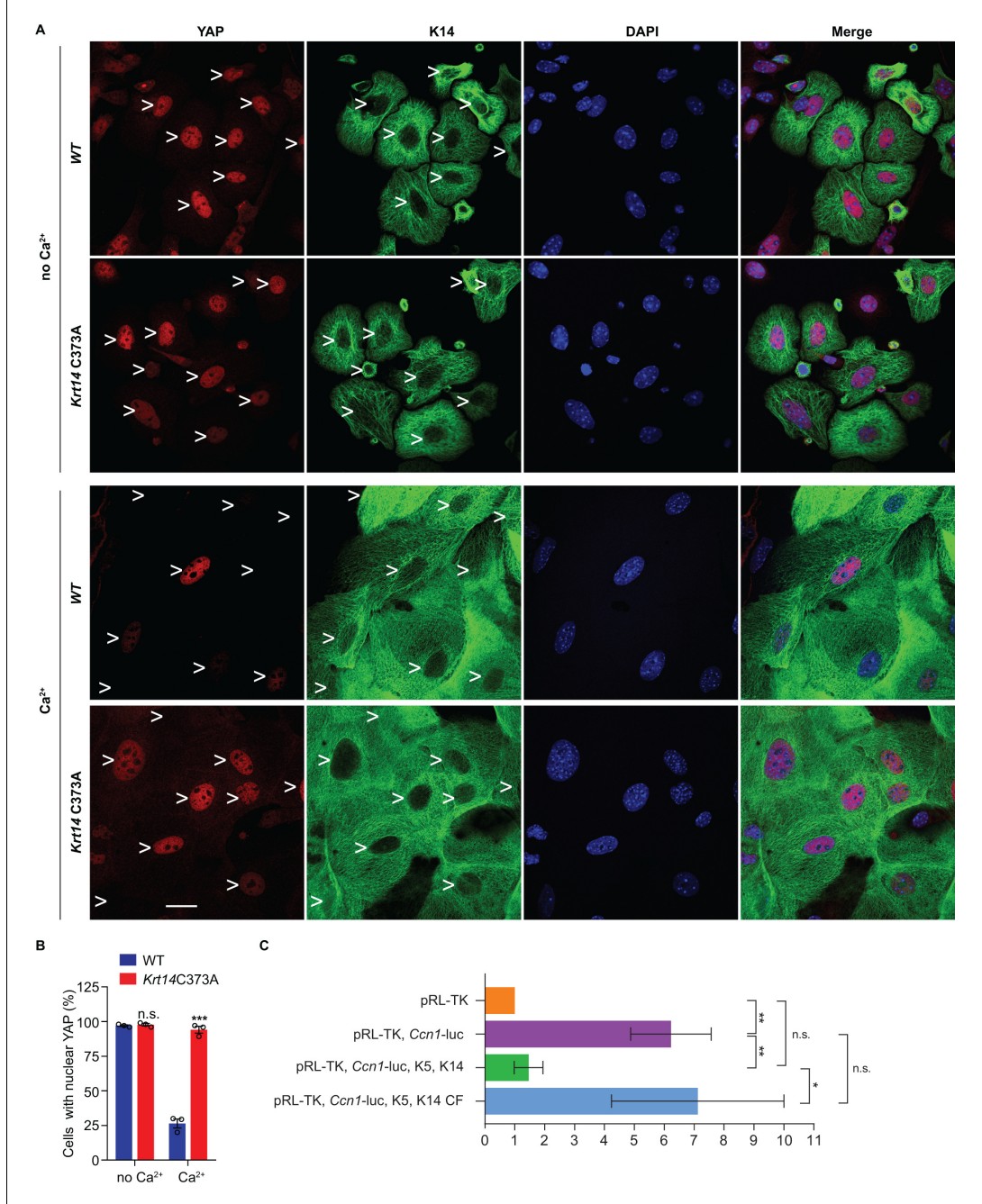

**Figure 5.** Localization and interaction of 14-3-3σ and YAP activity in *Krt14* C373A keratinocytes. (**A**) Indirect immunofluorescence microscopy for YAP (red) and K14 (green) in WT and *Krt14* C373A newborn skin keratinocytes in primary culture in the absence and presence of 1 mM calcium (for 4 d). Arrowheads depict location of nuclei. (**B**) Quantification of cells with nuclear YAP in frame a. N = 3 biological replicates. Approx. 100 cells were counted for each genotype for each condition. N = 3 biological replicates. Data represent mean ± SEM. Student's t test: n.s., no statistical difference; ***$p < 0.005$. Scale bars, 20 µm. (**C**) Luciferase assays in HeLa cells transfected with a *Ccn1*-Luciferase reporter construct (see Materials and methods). Data were normalized with regard to transfection efficiency and signal obtained with pRL-TK vector control. Data represent mean ± SEM three biological replicates consisting of 6 technical replicates each. Mann-Whitney tests were performed to compare each parameter using GraphPad Prism 8. **$p < 0.01$, *$p < 0.05$, n.s., non-significant.

The online version of this article includes the following figure supplement(s) for figure 5:

**Figure supplement 1.** Localization of YAP is specifically regulated by cysteine residue 367 in human K14.

differentiation. Moreover, PLA assays yielded evidence for decreased interactions between 14-3-3σ and YAP in *Krt14* C373A keratinocytes in primary culture, relative to WT controls (*Figure 4—figure supplement 1E,F*), thereby extending the immunoprecipitation findings shown in *Figure 4B*. We conclude that the abnormal retention of YAP to the nucleus and abnormal differentiation of *Krt14* C373A keratinocytes are both preserved outside of the skin tissue setting, further suggesting that these properties are linked and inherent to keratinocytes.

The five cysteine residues present in human K14 are conserved in the mouse ortholog (*Lee et al., 2012*), and cysteines at positions 4, 40, and 367 in human K14 participate in disulfide bonding (*Feng and Coulombe, 2015a*). We next asked whether the aberrant YAP localization in *Krt14* C373A mutant epidermis is Cys373-specific. To this end we devised a rescue assay using *Krt14* null mouse keratinocytes in primary culture, transfection of GFP-K14 or mutants thereof, and analysis of YAP subcellular partitioning. Consistent with previous findings (*Sambandam et al., 2015*), only 19.5% of GFP-K14WT-expressing keratinocytes exhibit nuclear YAP in the presence of calcium (*Figure 5—figure supplement 1A,B*). By contrast, cells expressing either a GFP-K14 cysteine free (CF) mutant, or a GFP-C367A single mutant, feature an abnormally high nuclear retention of YAP (83.8% and 81.9%, respectively) (*Figure 5—figure supplement 1A,B*). Restoring Cys367 in the K14-CF backbone (GFP-K14CF-C367 construct) rescued the abnormal nuclear retention of YAP, given that only 27.3% of transfected cells show YAP in the nucleus (*Figure 5—figure supplement 1A,B*). These findings directly implicate the stutter cysteine (C367 in human K14) as a calcium-dependent regulator of the subcellular partitioning of YAP in skin keratinocytes.

In an effort to directly relate K14 to the activity of YAP1 as a transcriptional regulator, we next conducted luciferase reporter assays in a heterologous cell culture setting. HeLa cells were selected because they are epithelia-derived, and grow and transfect well. HeLa cells normally show low levels of K14 expression (*Moll et al., 1982*) but respond well to K5-K14 co-transfection (*Figure 5—figure supplement 1C*). Transfection of a *Ccn1* (*Cyr61*) gene promoter-driven Firefly luciferase plasmid, previously shown to effectively report on YAP transcriptional activity (*Ma et al., 2017*), led to a strong (~8 fold) induction of luciferase activity normalized to cells transfected with a reference Renilla luciferase plasmid (*Figure 5C*). Consistent with K14's ability to retain YAP in the cytoplasm, *Ccn1* promoter activity was significantly attenuated by co-expression of WT K14 along with assembly partner WT K5 (*Figure 5C*). In striking contrast, activity of the *Ccn1* promoter construct was much less affected when cysteine-free K14 (K14 CF) was co-expressed with WT K5 (*Figure 5—figure supplement 1C*). Of note, the organization of filaments in HeLa cells co-expressing either K5 and K14WT or K5 and K14CF is comparable (*Figure 5—figure supplement 1C*). The findings from this heterologous assay substantiate the notion that K14-containing filaments impact YAP activity as predicted, and that this property depends on the presence of Cys residues in K14.

YAP is a key effector of mechanosensing and mechanotransduction (*Dupont et al., 2011*; *Benham-Pyle et al., 2015*; *Panciera et al., 2017*). Cells experiencing tension often respond by enhancing determinants such as F-actin stress fibers, acto-myosin contraction, the recruitment of α-catenin and vinculin to adherens junctions (*Leckband and de Rooij, 2014*; *Yap et al., 2018*), expression of lamin A/C in the nucleus (*Swift et al., 2013*), and frequency of binucleation (*Cao et al., 2017*). Monitoring the status of such elements provides a test for altered mechanosensing or mechanotransduction. In tail skin tissue sections, the immunostaining for α-catenin and for lamin A/C but not that for desmoglein 1 are markedly increased in *Krt14* C373A mice relative to WT controls (*Figure 6A,B*). Use of the a-18 antibody that recognizes a mechanosensitive epitope on α-catenin (*Shimoyama et al., 1992*) confirms that epidermal keratinocytes are under altered tensile stress in *Krt14* C373A epidermis (*Figure 6C*). In the setting of primary culture, we observed a greater incidence of multi-nucleated keratinocytes in $Ca^{2+}$-treated *Krt14* C373A newborn skin keratinocytes compared to WT control (14.3% versus 2.8%; *Figure 6D*). We also find that the immunofluorescence signal for α-catenin, lamin A/C, F-actin stress fibers and phosphorylated myosin light chain II (pMLC Ser19) are each increased in *Krt14* C373A relative to WT (*Figure 6E,F*). Collectively these findings strongly suggest that, consistent with YAP misregulation (see *Figure 5*), *Krt14* C373A mutant keratinocytes show alterations in mechanosensing and/or mechanotransduction relative to WT.

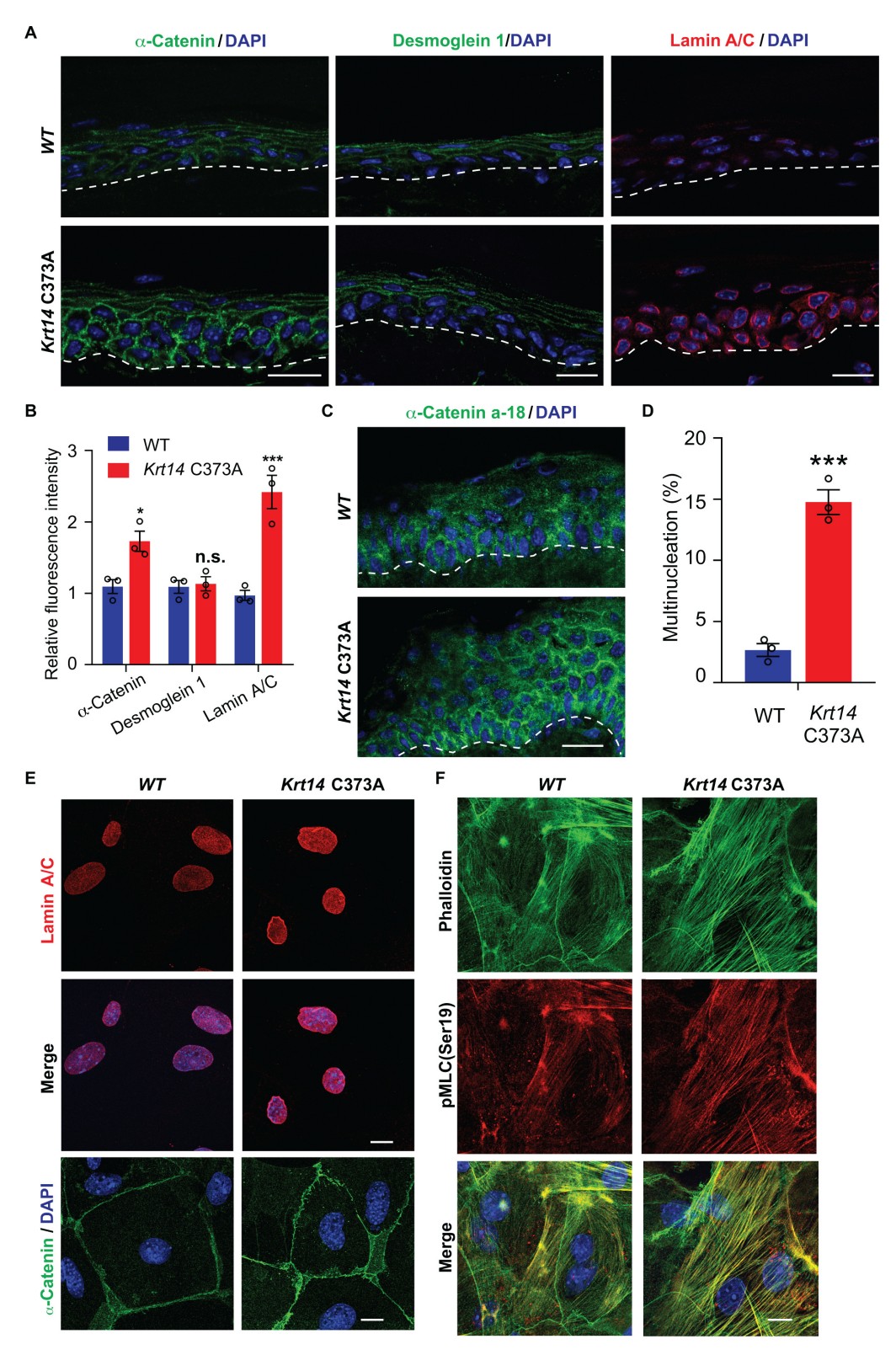

**Figure 6.** Altered mechanics in *Krt14* C373A epidermis and keratinocytes in culture. (**A-C**) Studies involving tail skin sections from young adult WT and *Krt14* C373A mice. A. Indirect immuno-fluorescence microscopy for α-catenin, desmoglein one and lamin A/C. Dashed lines depict the dermo-epidermal interface. B. Quantification of relative fluorescence intensity, as shown in frame a, for WT and *Krt14* C373A. N = 3 biological replicates. (**C**) Indirect immunofluorescence microscopy for the a-18 mechanosensitive epitope in α-catenin in tail skin sections from WT and *Krt14* C373A mice (see *Figure 6 continued on next page*

*Figure 6 continued*

A). D-F: Studies involving newborn skin keratinocytes in primary culture. (D) Percentage of cells with multinucleation in WT and *Krt14* C373A keratinocytes cultured as described for frames a,c. N = 3 biological replicates (total of 100 cells counted each time per genotype). (E) Indirect immunofluorescence microscopy for lamin A/C (top and middle rows) and α-catenin (bottom row) in primary cultures of WT and *Krt14* C373A newborn keratinocytes. (F) same as E, except that F-actin (via phalloidin) and Ser19-phosphorylated myosin light chain pMLC (Ser19) are stained. Nuclei are stained with DAPI in frames A, C, E and F. Scale bars, 20 μm. Data in B and F represent mean ± SEM. Student's t test: *p<0.05; ***p<0.005; n.s., no statistical difference.

## Discussion

Our study establishes that residue cysteine 373 in mouse K14 partakes in regulating the balance between keratinocyte proliferation and differentiation in epidermis in vivo, and ultimately barrier function, in skin. The loss of cysteine 373 results in profound alterations in the i) pattern of K14-dependent disulfide bonding in epidermis; ii) regulation of 14-3-3sigma and YAP in early-stage differentiating keratinocytes; and iii) several mechanosensitive readouts in the young adult epidermis in situ and newborn skin keratinocytes in primary culture. Our findings support the conclusion that K14-dependent disulfide bonding involving the conserved 'stutter cysteine' residue impacts cell architecture, mechanosensing and Hippo signaling at an early stage of epithelial differentiation in the epidermis.

A model that conveys the significance our findings is given in *Figure 7A*. Consistent with the literature, the model posits that Hippo signaling is inactive in most keratinocytes in the basal layer of epidermis (*Schlegelmilch et al., 2011*; *Beverdam et al., 2013*; *Sambandam et al., 2015*; *Totaro et al., 2017*), with YAP localizing to the nucleus given a specific level of cellular crowding and/or integrin-mediated adhesion to the extracellular matrix (*Panciera et al., 2017*; *Elbediwy and Thompson, 2018*). In basal keratinocytes, K5-K14 filaments are organized in loose bundles that run alongside the nucleus (*Coulombe et al., 1989*; *Lee et al., 2012*) while 14-3-3σ occurs at low levels (*Dellambra et al., 2000*; *Reichelt and Magin, 2002*; *Kim et al., 2006*). The first prediction of our model is that reception of differentiation-promoting cues triggers K14-dependent disulfide bonding and creates binding sites for 14-3-3 on K5-K14 filaments. The latter likely occurs via site-specific phosphorylation, on K14 (e.g., *Inaba et al., 2018*). These events foster the known reorganization of keratin filaments into a prominent perinuclear network (*Lee et al., 2012*), along with the binding and sequestration of YAP1 to the cytoplasm, thus activating Hippo signaling as keratinocytes initiate terminal differentiation (*Figure 7A*). Several reports converged in establishing a role for mitochondria-derived reactive oxygen species as a key trigger towards the initiation of terminal differentiation in keratinocytes of epidermis (*Tamiji et al., 2005*; *Bause et al., 2013*; *Hamanaka et al., 2013*; *Sun et al., 2015*). Such species could non-enzymatically mediate K14-dependent disulfide bonding in the perinuclear cytoplasm of keratinocytes (*Suzuki et al., 2019*).

Many type I keratins expressed in epidermis (e.g., K10, K16, K17) feature a cysteine residue at the location corresponding to codon 373 in mouse K14 and codon 367 in human K14 (*Lee et al., 2012*). Four lines of evidence support the contention that K10, in particular, is a strong candidate for K14-like regulation of YAP1 subcellular partitioning and Hippo signaling in the epidermis. First, *KRT10* (human) and *Krt10* (mouse) expression is turned on at the earliest stage of terminal differentiation (*Woodcock-Mitchell et al., 1982*; *Schweizer et al., 1984*), and the K10 protein features a structurally and biochemically equivalent Cys residue at position 401 and is capable of interacting with 14-3-3 (*Wilker et al., 2007*; *Huang et al., 2010*). Second, the crystal structure of the interacting 2B segments of K1-K10 (*Bunick and Milstone, 2017*) shows an overall fold identical to our original K5(2B)-K14(2B) structure (*Lee et al., 2012*), including the presence of a trans-dimer, homotypic disulfide bond mediated by the stutter cysteine (C401) in K10. Third, we previously showed that K10 partakes in the formation of disulfide-dependent, dimer-sized species in skin keratinocytes (*Lee et al., 2012*). Fourth the *Krt10* null mouse, described >15 years ago (*Reichelt and Magin, 2002*; *Reichelt et al., 2004*), exhibits a strong phenotype consisting of hyperproliferation, faster keratinocyte transit time, and impaired differentiation in the epidermis which, molecularly, correlates with a marked upregulation of 14-3-3sigma and c-Myc. *MYC* has since then been shown to be a *bona fide* YAP1 target gene (*Schütte et al., 2014*; *Kim et al., 2017*; *Cai et al., 2018*). Accordingly, our model (*Figure 7A*) predicts that the newly-defined role for K14 in regulating the subcellular partitioning of YAP and onset

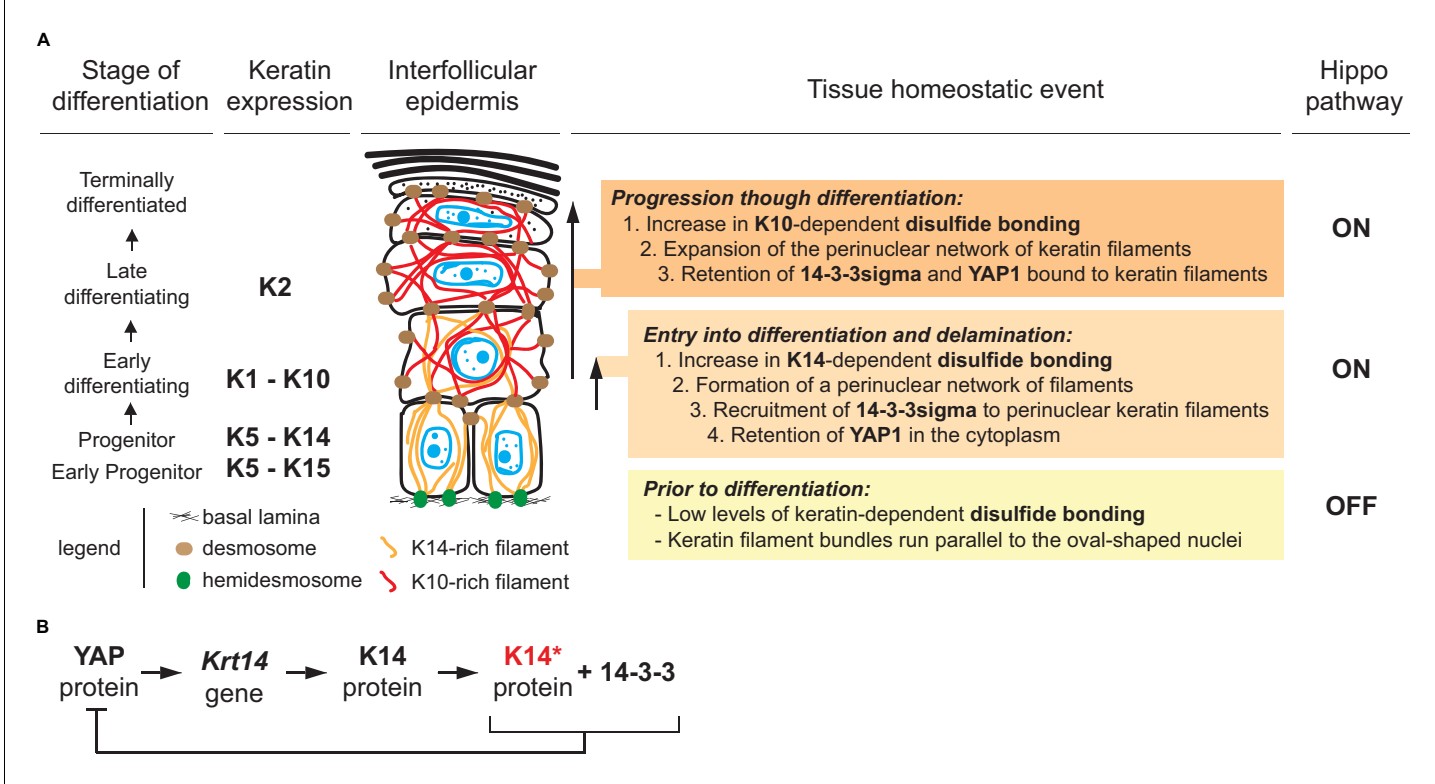

**Figure 7.** Model depicting the role of keratin-dependent disulfide bonding in integrating signaling and mechanical cues as keratinocytes initiate terminal differentiation in epidermis. (**A**) Left to right: the stage of epidermal differentiation, keratin expression, epidermal morphology, and state of keratin filament organization are related to 14-3-3 binding, YAP1 subcellular partitioning, and Hippo activity status. The model proposes that initiation of terminal differentiation in late stage progenitor keratinocytes in the basal layer entails: (1) the formation of K14-dependent disulfides via the conserved stutter cysteine in coil two domain; (2) a reorganization of keratin filaments around the nucleus; (3) recruitment of 14-3-3 onto keratin filaments; and (4) effective sequestration of YAP1 in the cytoplasm, resulting in activation of Hippo signaling. The model proposes an identical role for the conserved cysteine in coil 2 of keratin 10, which is expressed early during terminal differentiation, thereby maintaining YAP1's sequestration to the cytoplasm and active Hippo signaling. These changes are coupled to a redistribution of tension-related forces and cell-cell adhesion complexes as basal keratinocytes delaminate and move from the basal to the suprabasal compartment of epidermis (*Miroshnikova et al., 2018*; *Nekrasova et al., 2018*; *Wickström and Niessen, 2018*). (**B**) Illustration of a negative feedback loop whereby, once modified in a specific manner (phosphorylation and disulfide bonding), K14 protein sequesters YAP1 and interrupts its activity towards promoting keratinocyte proliferation, thereby initiating terminal differentiation (see *Figure 7—figure supplement 1* and 2 for related data).

The online version of this article includes the following figure supplement(s) for figure 7:

**Figure supplement 1.** Genomic context, gene expression, chromatin organization, and presence of TEAD binding sites in gene loci of interest.

of keratinocyte differentiation in epidermis is picked up and maintained by K10 as differentiation proceeds.

Our findings extend previous reports linking 14-3-3σ to the regulation of YAP during terminal differentiation in epidermis (*Sambandam et al., 2015*; *Sun et al., 2015*). They suggest that the network of keratin filaments proximal to the nucleus acts as a key docking site for 14-3-3/YAP complexes during keratinocyte delamination and differentiation, taking on a role that has been held to this point by integrin-based adhesion sites (*Elbediwy et al., 2016*) and adherens junctions (via alpha-catenin; (*Schlegelmilch et al., 2011*; *Silvis et al., 2011*). In addition, a disulfide bonding-rich perinuclear network of keratin filaments could also afford protection to the nucleus and the genome during delamination (*Lee et al., 2012*), thus extending the known role of K5-K14 IFs in providing mechanical support in basal keratinocytes (*Coulombe et al., 1991a*; *Ramms et al., 2013*). Besides, keratinocyte delamination and differentiation in epidermis together entail a profound reorganization of cell-cell and cell-matrix adhesion, and of F-actin and microtubule organization (*Sumigray and Lechler, 2015*; *Muroyama and Lechler, 2017*; *Miroshnikova et al., 2018*; *Rübsam et al., 2018*; *Wickström and*

*Niessen, 2018*). How all of these effectors and their inputs are integrated to result in a redistribution of intracellular tension and/or compressive forces (*Miroshnikova et al., 2018*; *Rübsam et al., 2018*), along with proper regulation of YAP function and Hippo signaling, awaits further investigation.

YAP's role as a transcription factor requires its binding to TEAD protein, which is itself stably bound to the promoter of its target genes in a sequence-specific fashion in the nucleus (*Vassilev et al., 2001*). Data available through ENCODE for human foreskin keratinocytes in primary culture in the absence or presence of calcium (see Materials and methods) provides insight into the rationale for keratin protein involvement towards the regulation of YAP-driven transcription and Hippo signaling. The combination of DNAse one hypersensitivity mapping, ATAC-seq mapping, and presence of TEAD binding sites (in either orientation) on DNA (5'- CATTCC-3'; *Heinz et al., 2010*) suggests that keratin genes expressed in dividing basal cells of epidermis, for example the type I *Krt14* and *Krt15* and type II *Krt5*, are likely YAP target genes given the presence of consensus-matching TEAD binding sites occurring in open chromatin regions in the absence of calcium (*Figure 7— figure supplement 1*; *Supplementary file 2*). ENCODE data convey that the chromatin surrounding these genes becomes non-accessible in the presence of calcium (*Figure 7—figure supplement 1* and 2), which promotes differentiation (*Hennings et al., 1980*). Keratin genes expressed in differentiating keratinocytes of epidermis, including the type I *Krt10* and type *Krt1* and *Krt2*, do not feature proximal TEAD binding sites (*Figure 7—figure supplement 1*; *Supplementary file 2*) and thus are not likely to be transcribed in a TEAD/YAP-dependent fashion in epidermis. The *YAP1 and TEAD* genes (human) are themselves poised to be positively regulated by YAP in progenitor keratinocytes (*Figure 7—figure supplement 1*; *Supplementary file 2*). *SFN*, *ITGB1*, *CCN1* and *TP63* (human) are among the additional genes of interest that we included in this analysis (*Supplementary file 2*). Accordingly, the YAP/TEAD axis may be set up to promote unabated cell proliferation in a fast-renewing epithelium such as epidermis until two post-translational events converging on K14 protein, phosphorylation and disulfide bonding, would lead to YAP protein sequestration in the cytoplasm and inactivation of the YAP/TEAD axis as part of a regulated negative feedback loop, enacted at the time of entry into differentiation in epidermis (*Figure 7B*).

Several mechanisms could account for the functional interplay between K14-dependent disulfide bonding, 14-3-3σ, and the regulation of YAP's subcellular partitioning in keratinocytes. First, the perinuclear enrichment of keratin IFs, which is promoted by K14-dependent disulfide bonding, is poised to increase the local concentration of binding sites for 14-3-3σ and YAP near the nucleus (simple mass action law). Second, the occurrence of K14-dependent disulfides may create an optimal binding interface for 14-3-3σ and YAP in the cytoplasm residing proximal to the nucleus. Third, the nucleus is known to function as a mechanosensor (*Cao et al., 2017*), and local forces impacted by the perinuclear network of keratin filaments could alter the mechanical gating of YAP across nuclear pores (*Elosegui-Artola et al., 2017*). These three mechanisms, and possibly others, could act in combination. How K14 (and possibly K10), 14-3-3σ, YAP, and other crucial effectors bind each other as part of this newly defined signaling axis, its regulation, and its significance have now emerged as open issues of high significance for future studies.

## Materials and methods

### Animals

All mouse studies were reviewed and approved by the Institutional Animal Use and Care Committee (IACUC) at both Johns Hopkins University and the University of Michigan. WT and *Krt14* C373A mice (C57BL/6 strain background) were maintained under specific pathogen-free conditions and fed rodent chow and water ad libitum. Male and female C57Bl/6 mice of 2–3 months of age (young adults) were used for all studies unless indicated otherwise. Mice were genotyped using standard PCR assays with oligonucleotides listed in *Supplementary file 3*.

### Generation of *Krt14* C373A mice using CRISPR-Cas9 technology

Krt14 C373A mice were generated using the RNA-guided CRISPR-Cas9 system as described (*Wang et al., 2013*). A guide RNA (gRNA) was selected and designed according to a gRNA CRISPR design tool (http://crispr.technology) (*Jaskula-Ranga and Zack, 2016*). Briefly, a Cas9 target site (GGGCCAGCTGCATGCAGTAACGG; with the PAM motif underlined) was selected based on having

a cut site proximal to codon C373 and low-predicted off-targets. Oligonucleotides were used to clone the target into pT7gRNA, and the plasmid was amplified and linearized prior to T7 transcription. The gRNA was transcribed in vitro and purified prior to injection. The homology directed repair (HDR) template was purchased as a 183-nt single stranded Ultramer (IDT), and encoded a TGC (Cys) to GCA (Ala) mutation at codon 373 of the mouse K14 coding sequence. The gRNA, Cas9 mRNA, and HDR template were co-injected into C57Bl/6 zygotes by the JHU Transgenic Core Facility. Potential transgenic founders were screened using restriction digestion of PCR product extending beyond the repair template oligonucleotide and findings were confirmed by direct DNA sequencing (data not shown). Several founders exhibited the desired recombination event, either as homozygotes or heterozygotes. Two male homozygotes founders were selected and independently backcrossed by mating to C57Bl/6 *wildtype* females for two generations to eliminate potential off-target effects. The *Krt14* C373A homozygotes used in this study were from *Krt14* C373A het x het breedings (for body weight measurements and epidermal thickness measurements) or hom x hom breedings (other experiments). The two lines analyzed exhibited consistent and identical phenotypes.

## Key reagents

A list of key reagents used in this study can be found in *Supplementary file 4* ('Key Resources Table File').

## Topical acetone treatments of *Krt14* C373A mice

The left ears of age-matched WT and *Krt14* C373A mice (2–3 months old) were topically treated with 40 µl acetone twice daily for 7 days (*Denda et al., 1996*). The volume of acetone applied was split equally between the dorsal and ventral sides of the ear. The right ear (same mice) was left untreated as control. Mice were anesthetized during acetone treatment as per IACUC standards. Immediately after the last treatment, mice were euthanized and tissue harvested for analysis.

## Transepidermal water loss (TEWL) measurements

Mice were anesthetized using isoflurane (delivered by inhalation) during TEWL measurements. Readings were obtained using a TEWAMETER TM300 (Courage and Khazaka, Köln, Germany) from adult WT and *Krt14* C373A mice at baseline and after the last topical treatment with acetone. Measurements were made from the dorsal side of ear skin. The TM300 probe was warmed for 2 min prior to each measurement, and held on the area of interest for a minimum of 30 reads until the alpha level was below 0.2, per the manufacturer's instructions.

## Measurement of cell proliferation through EdU labeling

EdU (A10044, Thermo Fisher Scientific) was prepared in PBS buffer at 10 mg/ml PBS and injected intraperitoneally into mice at a dose of 50 mg/kg body weight. Tail skin was harvested from anesthetized mice at 2 hr, 1 d, 3 d, and 7 d after injection and processed for immunofluorescence staining. EdU staining was performed using the Click-iT Plus EdU Alexa Fluor 488 Imaging Kit (catalog no. C10637, Thermo Fisher Scientific).

## Immunofluorescence staining of skin tissue sections

For indirect immunofluorescence staining, ear or tail samples were surgically harvested and immediately submerged into optimal cutting temperature (O.C.T.) media (25608–930, VWR Scientific), flash frozen on dry ice, and stored at –40°C until sectioning. 5 µm cryosections were cut in a specific and consistent tissue orientation in all experiments. Cryosections were allowed to thaw in PBS buffer at room temperature and incubated with primary antibodies followed by Alexa Fluor–conjugated secondary antibodies (Thermo Fisher Scientific), counterstained in DAPI (1;5, 000 in PBS; D1306, Thermo Fisher Scientific), and mounted in FluorSave Reagent mounting medium (345789, Calbiochem) for indirect immuno-fluorescence (*Hobbs et al., 2015*; *Kerns et al., 2016*). The primary antibodies used are listed in *Supplementary file 4* ('Key Resources Table File'). TUNEL staining for apoptotic cells was performed using the TUNEL enzyme (11767305001) and TUNEL label mix (11767291910) as recommended by the manufacturer (Roche Applied Science). Imaging was performed using either a Zeiss fluorescence microscope with an Apotome attachment or a Zeiss LSM 800 confocal microscope. All experimental and control preparations were imaged under identical

exposure conditions, and quantified using the ImageJ software (NIH) and ZEN lite 2.6 (ZEISS). Experimental data were collected from biological replicates (three or more) and technical replicates (typically two).

## Isolation and analysis of cornified envelopes (CEs)

CEs were isolated from dorsal ear and tail tissue from age-matched male WT and *Krt14* C373A mice. To separate dorsal from ventral ear tissue, we followed the Murine Skin Tissue Transplant protocol (*Garrod and D. Cahalan, 2008*). Extraction and preparation of CEs were performed using a protocol described by *Kumar et al., 2015*. Briefly, adult mouse ear skin or adult mouse tail skin (1 cm length) were boiled at 95° C (in place of hot water bath) for 20 min in 2 ml CE isolation buffer containing 20 mM Tris-HCl (pH 7.5), 5 mM EDTA, 10 mM dithiothreitol (DTT), and 2% sodium dodecyl sulfate (SDS). Half of the extracted sample (1 ml) was flash frozen and stored for future studies. CEs were extracted from the remaining (1 ml) portion of the CE isolate. Samples were centrifuged for 5 min at 5, 000 × g, rinsed in CE isolation buffer with 0.2% SDS, re-pelleted, resuspended in 250 µl of washing buffer, and stored at 4°C until seeded. For morphological evaluation, CE isolates from dorsal ear and tail skin were seeded on glass slides at a concentration of $1.5 \times 10^6$ CEs and $6 \times 10^6$ CEs, respectively, covered with a thin cover glass, and then imaged. CEs were isolated from four mice per genotype. Analysis of the area, circumstance, and aspect ratio (longest axis to the shortest axis) of CEs was performed using ImageJ software.

## Transmission electron microscopy

Ear tissue from 2 to 3 month old WT and *Krt14* C373A littermates was surgically harvested, minced, and fixed overnight at 4°C in 2% formaldehyde/2% glutaraldehyde in 0.1 M cacodylate buffer at pH 7.4. Samples were post-fixed in osmium tetroxide, counter-stained with uranyl acetate, and embedded in epoxy resin as previously described (*Lessard et al., 2013*). Thin sections were cut (50–70 nm thick), counter-stained with uranyl acetate and lead citrate, and examined using a Hitachi HU-12A transmission electron microscope. Toluidine blue- stained thick sections (1 µm thick) were used for morphological analyses at the light microscope level.

## RNA harvest, cDNA synthesis, and quantitative RT-PCR

RNA was harvested using TRIzol reagent (15596018, Thermo Fisher Scientific) and purified using the Nucleospin RNA kit (740955.250, Machery Nagel) according to the manufacturers' instructions. Concentration and purity for RNA samples were assessed by spectrophotometry. 1.0 µg RNA was reverse-transcribed with the iScript cDNA Synthesis kit (1708891BUN, Bio-Rad Laboratories) using the manufacturer's protocol. qRT-PCR was performed using iTaq Universal SYBR master mix (1725121, Bio-Rad Laboratories) on the CFX96 qRT-PCR apparatus (Bio-Rad Laboratories) as described (*Hobbs et al., 2015*; *Kerns et al., 2016*). The following program was used for all qRT-PCR reactions: denaturation step at 95°C for 5 min, 40 cycles of PCR (denaturation at 95°C for 10 s, annealing and elongation at 55°C for 30 s). No template or no reverse transcriptase controls, standard curves and a melt curve were included on every PCR plate. Normalized expression values from qRT-PCR data were calculated using Microsoft Excel by first averaging the relative expression for each target gene ($2^{-(\text{Cq target gene} - \text{Cq reference gene})}$) across all biological replicates and then dividing the relative expression value for the experimental condition by that for the control condition ($2^{-(\Delta\text{Cq experimental} - \Delta\text{Cq control})}$). Error bars were derived from the standard error of the mean (SEM) of the normalized expression values across all biological replicates. Normalized expression values for each target gene in all qRT-PCR experiments were derived from at least three biological replicates. Relative quantifications or fold changes of target mRNAs were calculated after normalization of cycle thresholds with respect to the reference gene β-actin. A list of all oligonucleotide primers used for target gene-specific custom qRT-PCR is provided in *Supplementary file 3*.

## Primary culture of skin keratinocytes and indirect immunofluorescence

Keratinocytes from 1 or 2 day old C57Bl/6 newborn mouse skin were isolated as described (*Wang et al., 2016*), and cultured in FAD medium (low calcium, 0.07 mM) for 1 day. Calcium switch experiments (*Wang et al., 2016*) were performed by switching to FAD medium supplemented with with 1 mM $CaCl_2$. Keratinocytes were harvested for analysis at 4 days or at 36 hr after calcium switch

as indicated in figure legends. For immunofluorescence, keratinocytes were fixed in 4% paraformal-dehyde (PFA), blocked in 10% normal goat serum/0.1% Triton X-100/PBS for 1 hr at room tempera-ture, incubated in primary antibody solution for 1 hr, washed in PBS, incubated in Alexa Fluor–conjugated secondary antibodies (Thermo Fisher Scientific), counterstained in DAPI (D1306, Thermo Fisher Scientific), and mounted in FluorSave Reagent mounting medium (345789, Calbiochem). Prox-imity ligation assay was performed according to the manufacturer's protocol (Duolink in Situ PLA, Sigma-Aldrich). F-actin was stained using the Alexa Fluor 488 Phalloidin (A123791, Thermo Fisher Scientific) according to the manufacturer's protocol. Micrographs were acquired using the Zeiss LSM 800 confocal microscope (Carl Zeiss Microscopy). Representative images from at least three inde-pendent experiments were shown. All images were and quantified by ImageJ software (NIH).

### Nucleofection of newborn mouse skin keratinocytes in primary culture

$Krt14^{-/-}$ skin keratinocytes (*Feng and Coulombe, 2015a*; *Feng and Coulombe, 2015b*) were cul-tured in FAD medium. pBK-CMV His-GFP-K14WT or cysteine variants (*Feng and Coulombe, 2015a*; *Feng and Coulombe, 2015b*) were transfected into $Krt14^{-/-}$ skin keratinocytes using P1 Primary Cell 4D-Nucleofector X Kit (V4XP-1024, Lonza). After nucleofection, cells were plated on collagen-coated coverglass and processed for analysis. For co-immunoprecipitation, HA-14-3-3σ (11946, Addgene) was transfected into skin keratinocytes in primary culture using the P1 Primary Cell 4D-Nucleofector X Kit (V4XP-1024, Lonza).

### Co-immunoprecipitation, protein gel electrophoresis, and mass spectrometry analysis

WT and $Krt14$ C373A keratinocytes in primary culture were washed with PBS and lysed in cold Triton lysis buffer supplemented with Empigen (1% Triton X-100; 2% Empigen; 40 mm Hepes, pH 7.5; 120 mm sodium chloride; 50μ MN-ethylmaleimide; 1 mm EDTA; 1 mm phenylmethyl-sulfonyl fluoride; 10 mm sodium pyrophosphate; 1 μg/ml each of chymostatin, leupeptin, and pepstatin; 10 μg/ml each of aprotinin and benzamidine; 2 μg/ml antipain; 1 mm sodium orthovanadate; and 50 mm sodium fluoride). Protein concentration was determined using the Bio-Rad protein assay (Bio-Rad Laborato-ries) with bovine serum albumin (Thermo Fisher Scientific) as a standard. For immunoprecipitation, aliquots of cell lysate were incubated with a K14 antibody, and immune complexes were captured using the Protein G Sepharose (17-0618-01, GE Healthcare). Samples for gel electrophoresis were prepared in Laemmli Sample Buffer (LDS) sample buffer (1610747, Bio-Rad) in the presence of 20 mM tris(2-carboxyethyl)-phosphine (TCEP) (77720, Thermo Fisher Scientific) and incubated at room temperature for 1 hr to reduce disulfide bonds. Non-reducing lysates were prepared directed in LDS sample buffer. Equal amounts of IP samples were resolved by 4–15% precast polyacrylamide gels (456–1084, Bio-Rad) and stained using a Silver Stain Kit (24612, Thermo Fisher Scientific). Bands of interest, along with a control area, were excised and analyzed by routine tandem mass spectrometry at the Johns Hopkins Mass Spectrometry Core. Mass spectrometry data were searched with Mascot 2.6.1 (Matrix Science) via Proteome Discoverer 2.2 (Thermo) against the RefSeq2017_83_mus_mus-culus Proteins database. Proteins with a false discovery rate (FDR) lower than 1% and with at least two identified peptides were reported as positive.

### Preparation of cell lysates, protein gel electrophoresis, and immunoblotting analysis

Cells or minced tissue were lysed in cold urea lysis buffer (pH 7.0, 6.5M urea, 50 mM Tris-HCl, 150 mM sodium chloride, 5 mM ethylenediaminetetraacetic acid (EDTA), 0.1% Triton X-100, 50 μM N-ethylmaleimide, 1 mM phenylmethanesulfonyl fluoride (PMSF), 1 μg/mL each of cymostatin, leu-peptin, and pepstatin, 10 μg/mL each of aprotinin and benzamidine, 2 μg/mL antipain, and 50 mM sodium fluoride). Protein concentration of the lysates was determined using Bradford protein assay (Bio-Rad) with bovine serum albumin as a standard. Samples for gel electrophoresis were prepared in LDS sample buffer (1610747, Bio-Rad) in the presence of 20 mM TCEP and incubated at room temperature for 1 hr to reduce disulfide bonds. Non-reduced lysates were prepared directed in LDS sample buffer. Equal amounts of cell lysates were resolved by 4–15% precast polyacrylamide gels (Bio-Rad) and transferred to nitrocellulose membrane (0.45 μm, Bio-Rad), and immunoblotted with the indicated antibodies followed by HRP-conjugated goat anti–mouse IgG or anti–rabbit IgG or

rabbit anti–chicken IgY (Sigma-Aldrich) and Super Signal West Pico Chemiluminescent Substrate (PI34080, Thermo Fisher Scientific) or Amersham ECL Select Western Blotting Detection Reagent (RPN2235, GE Healthcare). Signals were detected using the FluorChem Q imaging system (Protein Simple). The ImageJ software (NIH) was used for western blot signal quantitation.

## Luciferase assays

Luciferase assays were conducted in HeLa cells purchased from ATCC and authenticated using STR profiling (*Supplementary file 4*). These cells were tested routinely using a commercial luminescence assay (MycoAlert, Lonza) and found to be mycoplasma-free. Renilla luciferase control plasmid pRL-TK (Promega, E2241), YAP activity responsive Firefly luciferase plasmid *Cyr61*(*Ccn1*)-Luc (*Ma et al., 2017*), expression plasmid of human keratin 5 (K5), and expression plasmids of wildtype and cysteine-free (CF) keratin 14 (K14) (*Feng and Coulombe, 2015a*) were transfected into HeLa (ATCC) cells using SE Cell Line 4D X Nucleofector Kit S (V4XC-1032) with setting DS-138. After Nucleofection, cells were plated across six wells of a black matrix 96-well plate for each parameter. HeLa cells were transfected such that the cell density in each well was 30–40% the following morning. Firefly and Renilla luciferase activities were measured using Promega Dual Luciferase Reporter Assay System (Promega, PR-E1910). Firefly relative light unit (RLU) was normalized to internal Renilla RLU per well. Three biological replicates of normalized Firefly RLUs were pooled, and the means of each parameter were compared using a Mann-Whitney test. Data displayed were transformed by dividing individual RLUs of each parameter by the mean of pRL-TK alone and subjected to statistical analysis.

## Computational prediction of protein motifs

The predicted mouse K14 protein sequence (UniProtKB Q61781) was analyzed using publicly accessible algorithms written to predict 14-3-3 binding sites and phosphorylation events, including 14-3-3-Pred (*Madeira et al., 2015*) and Scansite 4.0 (*Obenauer et al., 2003*).

## ENCODE data

Data deposited in the ENCODE project were used to relate expression levels for genes of interest, chromatin accessibility in their proximal promoter region, and presence of TEAD binding sites. DNase-seq data from human newborn foreskin keratinocytes were produced by the Stamatoyannopoulos laboratory at University of Washington (Project: Roadmap, Award 01ES017156) and downloaded from the ENCODE portal (www.encodeproject.org) as a coverage file with the identifier ENCFF380PKB. RNA and ATAC sequencing data from male newborn human foreskin keratinocytes at 0, 3 and 6 days of calcium-induced differentiation were produced by the Greenleaf and Snyder laboratories at Stanford University (Project: GRR, Award: U01HG007919) and downloaded from the ENCODE portal. Coverage files of total RNA-seq data were downloaded with the following identifiers: ENCFF050SKD, ENCFF711YSO, ENCFF968JPE, ENCFF497JAC, ENCFF064QZN, ENCFF471GTD. Total RNA-seq data were loaded into the UCSC Genome Browser as bigwig files for visualization. Alignment files of ATAC-seq data were downloaded with the following identifiers: ENCFF111ULL, ENCFF654ZNI, ENCFF205KDV, ENCFF479UTZ, ENCFF374VWZ, ENCFF588PIS. For ATAC-seq data, alignment files were loaded into the Galaxy web platform (*Afgan et al., 2016*) using the public server at usegalaxy.org. Coverage files for visualizing ATAC-seq data were created using deepTools bamCoverage (*Ramírez et al., 2016*) with the following parameters: bin size = 5, normalization method = 1X (effective genome size GRCh38), smooth length = 10, exclude chrM for normalization, no extension. Regions of enriched ATAC signal were called using MACS2 callpeak (Galaxy Version 2.1.1.20160309.6; *Zhang et al., 2008*; *Feng et al., 2012*) on pooled replicates using the following parameters: `–format paired-end –gsize 2.7e9 –nomodel –qvalue` 0.05. Results from calling peaks on pooled replicates were loaded into the UCSC genome browser as narrowPeak files. To identify putative TEAD family binding sites, we loaded a file of genome-wide locations of predicted motifs from HOMER (*Heinz et al., 2010*) into the public Galaxy server and selected motifs corresponding to TEAD family members. These motifs were loaded as a custom track into the UCSC genome browser.

## Graphing and statistics

All graphs convey mean ± SEM values calculated using the Microsoft Excel software 2016 (Microsoft Office) or Prism software version 7 (GraphPad Software, Inc). For comparisons between datasets, the Student's t test (tails = 2) or Mann-Whitney tests were used, and statistically significant p-values are indicated in figures and figure legends.

## Acknowledgements

The authors are grateful to members of the Coulombe laboratory for support, Beau Su, Samuel Black and Younggook Oh for expert technical support, Steve Weiss and Yatrik Shah for reagents and advice, and Roger Reeves, Chip Hawkins and the Transgenic Core facility at the Johns Hopkins University School of Medicine for production of *Krt14* C373A mice. This research was supported by grant AR042047 to PAC from the National Institutes of Health.

## Additional information

### Funding

| Funder | Grant reference number | Author |
|---|---|---|
| National Institutes of Health | AR042047 | Pierre A Coulombe |
| National Institutes of Health | 5T32CA009676 | Catherine J Redmond |

The funders had no role in study design, data collection and interpretation, or the decision to submit the work for publication.

### Author contributions

Yajuan Guo, Data curation, Formal analysis, Validation, Investigation, Methodology; Catherine J Redmond, Conceptualization, Data curation, Formal analysis; Krystynne A Leacock, Investigation, Methodology; Margarita V Brovkina, Conceptualization, Resources, Data curation, Formal analysis; Suyun Ji, Formal analysis; Vinod Jaskula-Ranga, Conceptualization, Data curation, Methodology; Pierre A Coulombe, Conceptualization, Data curation, Formal analysis, Supervision, Funding acquisition, Investigation, Methodology

### Author ORCIDs

Catherine J Redmond https://orcid.org/0000-0001-9474-2803
Margarita V Brovkina https://orcid.org/0000-0002-5580-5872
Pierre A Coulombe https://orcid.org/0000-0003-0680-2373

### Ethics

Animal experimentation: This study was performed in strict accordance with the recommendations in the Guide for the Care and Use of Laboratory Animals of the National Institutes of Health. All of the animals were handled according to approved institutional animal care and use committee (IACUC) protocols of the University of Michigan. Every effort was made to minimize suffering.

### Decision letter and Author response

Decision letter https://doi.org/10.7554/eLife.53165.sa1
Author response https://doi.org/10.7554/eLife.53165.sa2

## Additional files

### Supplementary files

• Source data 1. Raw data used for quantitation.

• Supplementary file 1. Mass spectrometric screen for proteins interacting with wildtype K14 in WT newborn skin keratinocyte in primary culture. List of all proteins with at least 35 spectral counts (cumulative over three biological replicates). Gene symbols are provided at left. WT1, WT2, WT3, WT4 and WT5 represent different regions of silver stained protein electrophoretic gels subjected to MS analysis (data not shown). Keratin protein entries dominate this screen, as expected, and are listed using blue lettering. 14-3-3 protein isoforms are listed using red lettering. The top non-keratin protein from this screen is 14-3-3sigma.

• Supplementary file 2. Summary of data relating genomic context, gene expression, chromatin organization, and presence of TEAD binding sites in gene loci of interest. This summary account applies to human foreskin keratinocytes cultured in the absence (day 0) or presence of calcium for 6 days (see *Figure 7—figure supplement 1*) and is derived from the ENCODE project. Ratings for gene expression levels (low, moderate (med), high or very high) and promoter accessibility (low, moderate (med) or high) are reported for keratin genes known to be expressed in progenitor (*KRT14*, *KRT15*, *KRT5*) and differentiating keratinocytes (*KRT10*, *KRT1*, *KRT2*) in epidermis, and for additional genes that are relevant to our study (*SFN*, *ITGB1*, *CCN1*, *TP63*, *YAP1*, and *TEAD1*). The presence of consensus TEAD binding sites in the proximal promoter of these genes (either 1–2 sites, shown as '+", or >3 sites, shown as '+++"") is also reported. See Discussion for further information.

• Supplementary file 3. List of oligonucleotide primers used in this study.

• Supplementary file 4. Key Resources Table.

• Transparent reporting form

## Data availability

All of the data generated or analyzed during this study are included in the manuscript and supporting files.

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
