## [Decision Letter]

**Acceptance summary:**

Overall, this work provides important new insights into the role of cytoarchitecture in epidermal differentiation/homeostasis, and should be of broad interest to the *eLife* readership. They show that inhibiting the normal pattern of disulfide bonding in a single keratin species (keratin 14) is associated with effects on proliferation, transit time and differentiation in the epidermis. Further, they link this modification to the YAP binding partner 14-3-3 which impacts YAP signaling.

**Decision letter after peer review:**

Thank you for submitting your article "Keratin 14-dependent disulfides regulate epidermal homeostasis and barrier function via 14-3-3σ and YAP1" for consideration by *eLife*. Your article has been reviewed by two peer reviewers, and the evaluation has been overseen by a Reviewing Editor and Anna Akhmanova as the Senior Editor. The reviewers have opted to remain anonymous.

The reviewers have discussed the reviews with one another and the Reviewing Editor has drafted this decision to help you prepare a revised submission.

Summary:

Overall, this work provides important new insights into the role of cytoarchitecture in epidermal differentiation/homeostasis, and should be of broad interest to the readership if the comments of the reviewers are taken into consideration. Specifically, the authors show that inhibiting the normal pattern of disulfide bonding in this single keratin species is associated with effects on proliferation, transit time and differentiation. They identify a potential mechanism explaining these observations through identification of the YAP binding partner 14-3-3 as a K14 interacting protein. These data provide a new in vivo model for a fundamental mechanism integrating keratins, 14-3-3 and Hippo signaling (about which not that much is known in epidermis) during early stages of differentiation. Overall, the work identified an unexpected and important physiological role for keratin disulfide bridges in controlling epidermal morphology and function.

Essential revisions:

1) The reviewers felt that the analysis of mechanics (mostly data in Figure 6) were underdeveloped and gave a few suggestions for how this might be strengthened experimentally.

A) The data in Figure 6 is somewhat supportive of a mechanical function, the analysis of mechanics is somewhat superficial and indirect. Additional indicators would strengthen the analysis, including vinculin staining (vinculin associated with α-catenin increases under tension), a-18 antibody staining (recognizes epitope in α-catenin that is exposed when under tension), and more direct stiffness measurements.

B) Further, mechanistic connections between 14-3-3 association with IF, tension and YAP are not made in the paper. How are they related? In Figure 5C it is shown that 14-3-3 associates less well with YAP in context of mutant K14, suggesting that in WT cells, cytoplasmic 14-3-3, sitting on K14, keeps YAP out of the nucleus. Does this change in 14-3-3 cooperate with changes in mechanics to promote nuclear YAP, or alternatively (or in addition) is YAP driving the change in mechanics? WT cells appear to be under less tension, so this could correlate with reported drop in tension in cells destined to delaminate. But these connections are not made in the paper. The authors might want to consider using mutant forms of YAP to help establish the hierarchy downstream of cys mutant.

C) What are the respective contributions of each of these elements (disulfide keratin bonds/YAP signaling/14-3-3 binding) to observed alterations in differentiation? Asymmetries in mechanical properties in basal cells are important to drive delamination and promote stratification. So the decreased tension in WT cells driven by disulfide bonding of keratins could be a key player here.

2) The connection between disulfide bond formation and other possible effects of mutation of the Cys residue would be greatly strengthened by including analysis of backskin tissue, where formation of these disulfide bonds does not occur. What happens to the thickness, proliferation and YAP/14-3-3 localization? If these are all normal, it bolsters the connection between the mutation and the disulfide bonding. If the phenotypes are also present in backskin, this would complicate the conclusions.

---

## [Author Response]

Essential revisions:1) The reviewers felt that the analysis of mechanics (mostly data in Figure 6) were underdeveloped and gave a few suggestions for how this might be strengthened experimentally.A) The data in Figure 6 is somewhat supportive of a mechanical function, the analysis of mechanics is somewhat superficial and indirect. Additional indicators would strengthen the analysis, including vinculin staining (vinculin associated with α-catenin increases under tension), a-18 antibody staining (recognizes epitope in α-catenin that is exposed when under tension), and more direct stiffness measurements.

We have performed stainings of skin tissue sections using antibodies directed at vinculin and the α-catenin a-18 (mechanically-sensitive) epitope. The latter yielded compelling results that have been added to Figure 6. Otherwise, Figure 6 reports clear findings from both tissue sections and keratinocytes in primary culture using several mechano-sensitive readouts. At this point we do not have direct mechanical measurements comparing WT and *Krt14* C373A skin keratinocytes. We have cultivated newborn skin keratinocytes on matrices that exhibit a range of stiffness (8, 25, 50 and 100 kPa) and have made two promising observations: (1) the yield of K14-dependent disulfide-bonded species shows a dependency on ECM stiffness for *WT* cells, and (2) this relationship is different for *Krt14* C373A keratinocytes. These results are preliminary at this point. Getting them to a publication-quality level will require significantly more effort and time.

B) Further, mechanistic connections between 14-3-3 association with IF, tension and YAP are not made in the paper. How are they related? In Figure 5C it is shown that 14-3-3 associates less well with YAP in context of mutant K14, suggesting that in WT cells, cytoplasmic 14-3-3, sitting on K14, keeps YAP out of the nucleus. Does this change in 14-3-3 cooperate with changes in mechanics to promote nuclear YAP, or alternatively (or in addition) is YAP driving the change in mechanics? WT cells appear to be under less tension, so this could correlate with reported drop in tension in cells destined to delaminate. But these connections are not made in the paper. The authors might want to consider using mutant forms of YAP to help establish the hierarchy downstream of cys mutant.

These are important issues. The molecular reagents and assays available to us at present do not allow more resolution in our mechanistic understanding of the interplay between K14, K14-dependent disulfide bonding, YAP1, mechanosensing and mechanotransduction. We are being candid about open issues of importance in the Discussion. Besides, the editors and reviewers should agree that much remains to be learned about the distribution and evolution of forces, on various scales, as progenitor keratinocytes initiate terminal differentiation and delaminate to enter the suprabasal compartment.

C) What are the respective contributions of each of these elements (disulfide keratin bonds/YAP signaling/14-3-3 binding) to observed alterations in differentiation? Asymmetries in mechanical properties in basal cells are important to drive delamination and promote stratification. So the decreased tension in WT cells driven by disulfide bonding of keratins could be a key player here.

We agree with the notion that altered tension in normal keratinocytes, driven by keratin-dependent disulfide bonding, is a key player in driving delamination and promoting stratification (this notion is an intrinsic part of the model we propose in Figure 7). For reasons outlined above already, we cannot discriminate about the respective contributions of keratin disulfide bonding, 14-3-3 binding, and YAP signaling to these processes. Our data clearly establish that all three elements work in concert in this setting, with keratin dependent-disulfide bonding and the resulting impact on YAP subcellular partitioning and activity revealed as both a novel and unexpected determinant of these processes. Ours is the first evidence that keratin proteins are directly involved in the regulation of keratinocyte differentiation in epidermis in situ.

2) The connection between disulfide bond formation and other possible effects of mutation of the Cys residue would be greatly strengthened by including analysis of backskin tissue, where formation of these disulfide bonds does not occur. What happens to the thickness, proliferation and YAP/14-3-3 localization? If these are all normal, it bolsters the connection between the mutation and the disulfide bonding. If the phenotypes are also present in backskin, this would complicate the conclusions.

We added data (see Figure 3—figure supplement 1, entirely new) showing that there is no evidence of anomaly in the back skin of young adult *Krt14* C373A back skin tissue when assessing, for instance, epidermal thickness, markers of terminal epidermal differentiation, and Yap staining. This new data is consistent with the reviewer’s prediction and with the observation that there is considerably less evidence for K14-dependent disulfide bonding in adult back skin tissue as reported in Figure 1C.